



**Expansion of oil palm and other cash crops causes an increase of land surface temperature**
**in Indonesia**
Clifton R. Sabajo[1,2,†], Guerric le Maire[3], Tania June[4], Ana Meijide[1], Olivier Roupsard[3,5],
Alexander Knohl[1,6]
[1] University of Goettingen, Bioclimatology, 37077 Göttingen, Germany
[2] AgroParisTech – Centre de Montpellier, Agropolis International, 648 rue Jean-François
Breton, 34093 Montpellier, France
[3] CIRAD, UMR Eco&Sols, F-34398 Montpellier, France
[4] Agrometeorology Laboratory Department of Geophysics and Meteorology,
Faculty of Mathematics and Natural Sciences, Bogor Agricultural University (IPB), Indonesia
[5] CATIE (Centro Agronómico Tropical de Investigación y Enseñanza / Tropical Agriculture
Centre for Research and Higher Education), 7170 Turrialba, Costa Rica
[6] University of Goettingen, Centre of Biodiversity and Sustainable Land Use (CBL), 37073
Goettingen, Germany
[†] Correspondence: Clifton R. Sabajo, University of Goettingen, Bioclimatology, Büsgenweg 2,
37077 Göttingen, Germany. E-mail: csabajo@uni-goettingen.de
Telephone: +49 (0) 551 39 12114
**Abstract**
Indonesia is currently one of the regions with the highest transformation rate of the land surface
worldwide due to the expansion of oil palm plantations and other cash crops replacing forests
on large scales. Land cover changes, which modify land surface properties, have a direct effect



on the land surface temperature (LST), a key driver for many ecological functions. Despite the
large historic land transformation in Indonesia toward oil palm and other cash crops and
governmental plans for future expansion, this is the first study so far to quantify the impact of
land transformation in Indonesia on LST. We analyse LST from the thermal band of a Landsat
image and produce a high resolution surface temperature map (30m) for the lowlands of the
Jambi province on Sumatra (Indonesia), a region of large land transformation towards oil palm
and other cash crops over the past decades. We compare LST, albedo, Normalized Differenced
Vegetation Index (NDVI), and evapotranspiration (ET) of seven different land cover types
(forest, urban areas, clear cut land, young and mature oil palm plantations, acacia and rubber
plantations) and show that forests have lower surface temperatures than these land cover types
indicating a local warming effect after forest conversion with LST differences up to $10.09 \pm 2.6$
ºC (mean ± SD) between forest and clear cut land. The differences in surface temperatures are
explained by an evaporative cooling effect offsetting an albedo warming effect. Our analysis of
the LST trend of the past 16 years based on MODIS data shows that the average daytime surface
temperature of the Jambi province increased by 1.05 ºC, which followed the trend of observed
land cover changes and exceed the effects of climate warming. Our study provides evidence
that the expansion of oil palm plantations and other cash crops leads to changes in biophysical
variables, warming the land surface and thus enhancing the increase in air temperature due to
climate change.


*Keywords*: Land surface temperature, albedo, NDVI, evapotranspiration, biophysical variables,
oil palm, remote sensing, Landsat, MODIS, Indonesia, land-use / land cover change




## 1 Introduction

Indonesia is one of the regions where the expansion of cash crop monocultures such as acacia (timber plantation), rubber, oil palm plantations and smallholder agriculture has drastically reduced the area of primary forest in the past decades (Bridhikitti and Overcamp, 2012; Drescher et al., 2016; Marlier et al., 2015; Miettinen et al., 2012; Verstraeten et al., 2005). This large scale conversion of rainforest for agricultural use has been observed on the island of Sumatra, which has experienced the highest primary rainforest cover loss in all of Indonesia (Drescher et al., 2016; Margono et al., 2012; Miettinen et al., 2011). Forest cover in the Sumatran provinces of Riau, North Sumatra and Jambi, declined from 93 to 38% of provincial area between 1977 and 2009 (Miettinen et al., 2012). These large scale transformations, observed as land cover change, and land-use intensification have led to substantial losses in animal and plant diversity, and ecosystem functions and changed microclimatic conditions (Clough et al., 2016; Dislich et al., 2016; Drescher et al., 2016). Additionally, these changes directly alter vegetation cover and structure as well as land surface properties such as albedo, emissivity and surface roughness which affect gas and energy exchange processes between the land surface and the atmosphere (Bright et al., 2015).

Replacing natural vegetation with another land cover modifies the surface albedo, which affects the amount of solar radiation that is absorbed or reflected and consequently alters net radiation and local surface energy balance. A low or high albedo results in smaller or greater reflection of shortwave radiation. As a result the higher or lower amounts of net radiation absorption may rise or lower the surface temperature and change evapotranspiration (Mahmood et al., 2014).

Changes in land cover also alter surface emissivity, i.e. the ratio of radiation emitted from a surface to the radiation emitted from an ideal black body at the same temperature following the





Stefan–Boltzmann law. Emissivity of vegetated surfaces varies with plant species, density,
growth stage, water content and surface roughness (Snyder et al., 1998; Weng et al., 2004). A
change of emissivity affects the net radiation because it determines the emission of longwave
radiation that contributes to radiative cooling (Mahmood et al., 2014).

Water availability, surface type, soil humidity, local atmospheric and surface conditions affect
the energy partitioning into latent (LE), sensible (H) and ground heat (G) fluxes (Mildrexler et
al., 2011). Surface roughness affect the transferred sensible and latent heat by regulating vertical
mixing of air in the surface layer (van Leeuwen et al., 2011) thereby regulating land surface
temperature (LST). Through its association with microclimate, net radiation and energy
exchange (Coll et al., 2009; Sobrino et al., 2006; Voogt and Oke, 1998; Weng, 2009; Zhou and
Wang, 2011) LST is a major land surface parameter that also influences habitat quality and thus
the distribution of plants and animals and biodiversity.

The replacement of natural vegetation also changes evapotranspiration (ET) (Boisier et al.,
2014). In case ET is decreased, surface temperatures and fluxes of sensible heat (H) increase.
*Vice versa* when ET increases, increased LE fluxes lower surface temperatures and decrease H
fluxes (Mahmood et al., 2014). Vegetation structure as reflected by parameters such the
Normalized Difference Vegetation Index (NDVI), Leaf Area Index (LAI) and vegetation height
is in this respect an important determinant of the resistances or conductivities to heat, moisture,
and momentum transfer between the canopy and the atmosphere (Bright et al., 2015) facilitating
the amounts/ratios of sensible heat to water vapour dissipation away from the surface
(Hoffmann and Jackson, 2000).

Surface albedo, surface temperature, surface emissivity, and indirectly LAI and NDVI are
interconnected through the surface radiation balance. When the land surface is changed



feedback mechanisms involving these biophysical variables control the radiation balance and
the surface temperature.
To understand the effects of land cover changes on LST, the associated biophysical variables
must be evaluated. This can be done through the surface radiation budget and energy
partitioning which unites these biophysical variables directly or indirectly: albedo as direct
determinant of the net solar radiation, NDVI as a vegetation parameter determining the
emissivity which in turn determines the amount of reflected and emitted longwave radiation,
LST directly affecting the amount of emitted longwave radiation from the surface and ET
affecting the amount of energy that is used for surface cooling via evaporating of water.

The effect of land cover change on LST is dependent on the scale, location, direction and type
of the change (Longobardi et al., 2016).  Several studies showed an increase of the LST after
forest were converted: in China built-up areas and  agricultural land (Zhou and Wang, 2011),
and in crop land and pasture lands (Peng et al., 2014). Similar findings were reported for South
American ecosystems: low vegetation such as grasslands in Argentina were warmer than tall
tree vegetation (Nosetto et al., 2005). In Brazil, the surface temperature increased after the
conversion of natural Cerrado vegetation (a savanna ecosystem) into crop/pasture (Loarie et al.,
2011a). Similar effects were also shown for other South American biomes (Salazar et al., 2016).
In a global analysis, Li et al. (2015) showed that the cooling of forests is moderate at mid
latitudes and that Northern boreal forests are even warmer, an indication that the effect of land
cover change on LST varies with the location of the land cover change (Longobardi et al.,
2016). Similar studies on the Indonesian Islands are lacking but increases in surface temperature
are expected as an effect of the expansion of oil palm and cash crop land in the recent decades.

Measuring changes in LST is critical for understanding the effects of land cover changes, but
challenging. LST can be monitored with LST products retrieved from thermal infrared (TIR)



remote sensing data e.g. the use of the thermal bands of the Moderate Resolution Imaging
Spectrometer (MODIS) onboard the Terra and Aqua satellite (Sobrino et al., 2008), the thermal
band of the Thematic Mapper (TM) onboard the LANDSAT-5 platform (Sobrino et al., 2004,
2008) or Enhanced Thematic Mapper (ETM+) onboard the LANDSAT-7 platform. The
advantage of MODIS data is the availability of readily processed products at high temporal
resolution (daily) at medium (250 – 500 m) to coarse spatial resolution (1000 – 5000 m) scale;
MODIS LST product (MOD11A1/MYD11A1) for example is provided at a daily temporal
resolution with a spatial resolution of 1 km. Landsat data are provided at a higher spatial
resolution (30 m), but its temporal resolution is however limited to 16 days and the retrieval of
LST requires the correction of the satellite observed radiances for atmospheric absorption and
emission (Coll et al., 2009). Besides LST, the connected biophysical variables of the energy
and radiation budget can be derived from the visible and near-infrared (VIS-NIR) bands of
either MODIS or Landsat, making integrated monitoring of the biophysical variables related to
changing land surface possible. In Indonesia, a large proportion of the land use changes is
driven by small holders (Dislich et al. 2016), thus a combination of Landsat (for a fine spatial
resolution) and MODIS (for temporal developments) seems desirable.

The modification of the physical properties of the land surface influences climate/local
microclimatic conditions via biogeochemical and biophysical processes. Therefore, given
Indonesia's history of large scale agricultural land conversion and governmental plans to
substantially expand the oil palm production, it is important to study the effect of the expansion
of cash crop areas on the biophysical environment, especially on LST as a key land surface
parameter. These effects have been poorly studied in this region and according to our
knowledge this is the first study to quantify the effects of land use change on LST in Indonesia
We focus on the province of Jambi / Sumatra as it experience large land transformation towards



oil palm and other cash crops such as rubber plantations in the past and may serve as example
of future changes in other regions.

Our main objective is to quantify the differences in LST across different land cover types and
to assess the impact of cash crop expansion on the surface temperature of Jambi province (on
Sumatra / Indonesia) in the past decades. With this study we aim to (1) evaluate the use of
Landsat and MODIS satellite data as sources for a reliable estimation of the surface temperature
in a tropical region with limited satellite data coverage by comparing the surface temperatures
retrieved from both satellite sources to each other and against ground observations, (2) to
quantify the LST variability across different land cover types  and (3) the long term effects of
land transformation on the surface temperature against the background of climatic changes and
(4) to identify the mechanisms that explain changes of the surface temperature through changes
in other biophysical variables. In this study we compare the surface temperatures of different
land cover types that replace forests (i.e. oil palm, rubber and acacia plantations, clear cut land
and urban areas) using high resolution Landsat and medium resolution MODIS satellite data
and discuss the differences by taking into account other biophysical variables such as the
albedo, NDVI and evapotranspiration (ET).

**2 Materials and methods**

**2.1 Study area**

The study was carried out in the lowlands (approx. 25 000 km$^2$) of the Jambi province (total
area 50 160 km$^2$) on Sumatra, Indonesia, between latitudes 0°30′S and 2°30′S and longitudes
101°E and 104°30′E (Fig. 1). This region has undergone large land transformation towards oil
palm and rubber plantation over the past decades and thus may serve as an example of expected



changes in other regions of Indonesia (Drescher et al. 2016). The area has a humid tropical
climate with a mean annual temperature of 26.7 ± 0.2 °C (1991 – 2011, annual mean ± SD of
the annual mean), with little intra-annual variation. Mean annual precipitation was 2235 ± 381
mm and a dry season with less than 120 mm monthly precipitation usually occurred between
June and September (Drescher et al., 2016). Details about the study area can be found in
(Drescher et al., 2016).

For this study, we used two data sets of different plot sizes. For the first data set, we delineated
28 large plots (ranging from 4 to 84 km$^2$) of 7 different land cover types (Forest (FO), Rubber
(RU), Acacia Plantation Forest (PF), Young oil palm plantation (YOP), Mature Oil Palm
Plantation (MOP), Urban area (UB) and Clear Cut areas (CLC)) (Fig. 1). The delineation was
based on visual interpretation in combination with information from field work, which was
carried out between October – December 2013. The large size of the plots was necessary to
make a comparison between MODIS and Landsat images (see section satellite data). For the
second data set, we selected within and outside these 28 large plots 49 smaller plots (between
50 × 50 m and 1000 × 1000 m) (Fig. 1) which allowed us to increase the number of plots to use
when analysing Landsat images. These small plots were used to extract surface temperature
(LST), Normalized Difference Vegetation Index (NDVI), albedo (α) and evapotranspiration
(ET) from a high resolution Landsat satellite image (see section satellite data) for the 7 different
land cover types of interest.



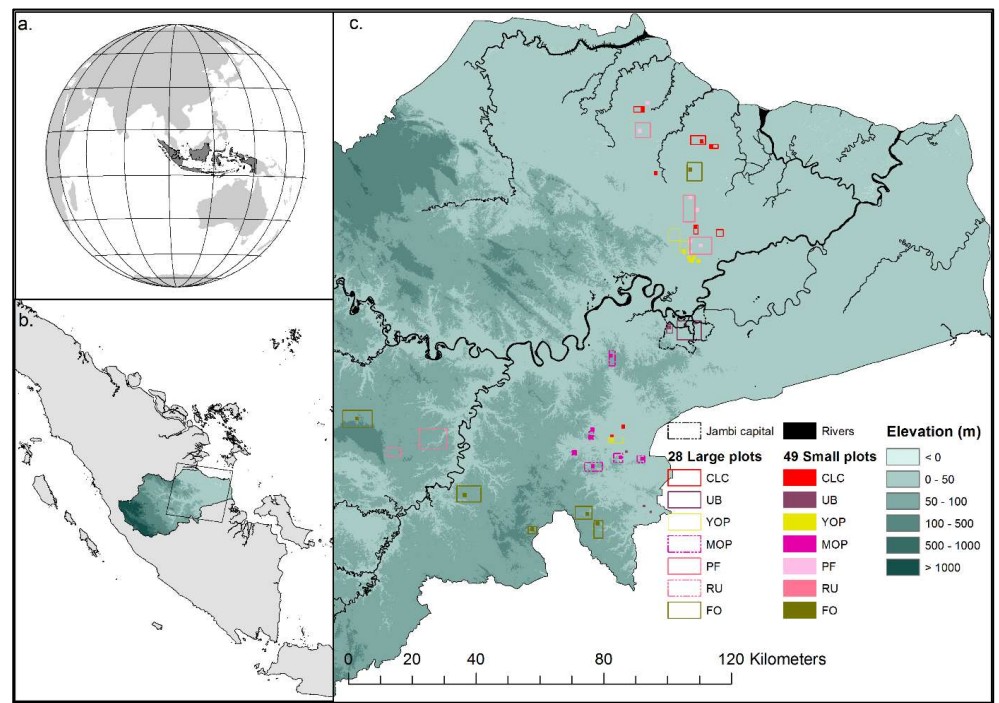

**Fig. 1** Geographic location of the study area. Jambi province on the Sumatran Island of
Indonesia (Figs. 1a and 1b). The background of the map (Fig. 1c) is a digital elevation model,
showing that the plots are located in the lowlands of the Jambi province. The large rectangles
are the 28 different land cover types (Forest, Young and Mature Oil palm, Rubber, Urban area,
Acacia Plantation Forest and Clear Cut land), the small squares are the locations of the 49 small
plots of the 7 different land cover types. Abbreviations: CLC = Clear cut land, UB = Urban
area, YOP = Young oil palm plantation, MOP = Mature Oil Palm plantation, PF = Acacia
plantation forest, RU = Rubber plantation, FO = Forest.

**2.2 Meteorological data**

Air temperature and relative air humidity were measured at four reference meteorological
stations located in open areas within the area of study (Drescher et al., 2016), with
thermohygrometers (type 1.1025.55.000, Thies Clima, Göttingen, Germany) placed at 2m
height. Measurements were taken every 15 s and then averaged and stored in a DL16 Pro data



logger (Thies Clima, Göttingen, Germany) as 10 min mean, from February 2013 to December
2015. We used the air temperature from the meteorological stations to compare to MODIS air
temperatures (MOD07_L2). The relative air humidity was used as an input parameter for
NASA's online atmospheric correction (ATCOR) parameter tool to derive parameters to correct
Landsat thermal band for atmospheric effects (see Satellite data). We also used air temperature
and relative humidity from two eddy covariance flux towers located in the study area (Meijide
et al., 2017) one in a young oil palm plantation (two years old, S 01°50.127', E 103°17.737'),
and the other one in a mature oil palm plantation (twelve years old, S 01°41.584', E
103°23.484'). At these flux towers, air temperature and relative humidity were measured above
the canopy respectively with the same instruments as in the reference meteorological stations
(see Meijide et al. (2017), for description of methodology). In the flux tower located in the
mature oil palm plantation, we also measured surface canopy temperature between August 2014
and December 2015, which was compared to MODIS LST estimates from the same period.
Measurements of canopy temperature were performed with two infrared sensors (IR100)
connected to a data logger, (CR3000) both from Campbell Scientific Inc. (Logan, USA). For a
regional    coverage    we    used    ERA    Interim    daily    air    temperature    grids
(http://apps.ecmwf.int/datasets/data/interim-full-daily/levtype=sfc/; (Dee et al., 2011) from
2000 – 2015 at 0.125 degrees resolution to study the annual air temperature trend in this period.

**2.3 Satellite data**

A Landsat 7 ETM+ VIS/TIR 30 m resolution surface reflectance image with low cloud cover,
acquired at 10:13 hours (local time) on 19 June 2013 covering the lowland area of the Jambi
province (path 125, row 61) was used in this study. Like all Landsat 7 ETM+ images acquired
after 31 may 2003, the image we used was affected by a scan line error causing a data loss of
about 22% (http://landsat.usgs.gov/products_slcoffbackground.php). Most selected plots were



located in the center of the image and thus not affected by the data loss, e.g. the forest plots
located at the edges of the scan line error zone faced minimal data loss because they were large
enough.
We also downloaded the tile h28v09 of the MODIS Terra (MOD) and Aqua (MYD) daily 1km
Land Surface Temperature and Emissivity products (MOD11A1 and MYD11A1 Collection-5)
and MODIS 16-days 500 m Vegetation Indices NDVI/EVI product (MOD13A1 Collection-5)
from 05 March 2000 till 31 December 2015 for Terra data and from 8 July 2002 till 31
December 2015 for Aqua data. We downloaded other supporting satellite data such as the
MODIS Atmospheric Profile product (MOD07_L2) and the MODIS Geolocation product
(MOD03). All MODIS data were reprojected to WGS84, UTM zone 48 South using the MODIS
Reprojection Tool (MRT). The quality of the MODIS data was checked using the provided
quality flags and only pixels with the highest quality flag were used in the analysis.

**2.4 Retrieval of biophysical variables from Landsat 7 ETM+ VIS/TIR images**


• *NDVI*

NDVI was derived using the reflectances corrected for atmospheric effects in the red ($\rho$RED,
band 3 Landsat 7 ETM+) and near infrared ($\rho$NIR, band 4 Landsat 7 ETM+) bands, with:


$NDVI = \dfrac{\rho\text{NIR} - \rho\text{RED}}{\rho\text{NIR} + \rho\text{RED}}$ (1)





- *Surface albedo*

The surface albedo ($\alpha$) was computed using the equation of Liang (2000) for estimating broadband albedo from Landsat surface reflectance bands, with:

$$\alpha = 0.3141\,\rho1 + 0.1607\,\rho3 + 0.369\,\rho4 + 0.1160\,\rho5 + 0.0456\,\rho7 - 0.0057 \qquad (2)$$

where $\rho_1$, $\rho_3$, $\rho_4$, $\rho_5$ and $\rho_7$ are the Landsat 7 ETM+ surface reflectance bands (corrected for atmospheric effects).

- *Surface temperature (LST)*

LST was derived following the method proposed by Bastiaanssen (2000), Bastiaanssen et al. (1998a), Coll et al. (2010) and Wukelic et al. (1989) for computing the surface temperature from the thermal infrared band (TIR, band 6) of Landsat (Supporting information, S1). The thermal infrared band (TIR, band 6) was first converted to thermal radiance (L6, $W/m^2/sr/\mu m$) and then to atmospherically corrected thermal radiance (Rc, $W/m^2/sr/\mu m$) following the method described by Wukelic et al. (1989) and Coll et al. (2010), and using the atmospheric parameters obtained on NASA's online Atmospheric Correction Calculator (Barsi et al., 2003, 2005) (supporting information, S2). The surface temperature (LST, °K) was computed through the following equation similar to the Planck equation, as in Coll et al. (2010) and Wukelic et al. (1989):





$\text{LST} = \dfrac{k2}{\ln\left(\frac{\varepsilon NB \cdot k1}{Rc} + 1\right)}$                                                          (3)

where εNB is the emissivity of the surface obtained from the NDVI (Supporting information,
Table S1), k1 (= 666.09 mW/cm$^2$/sr/μm) and k2 (= 1282.71 °K) are sensor constants for
converting the thermal radiance obtained from band 6 of Landsat 7 to surface temperature.
The surface temperature derived from Landsat thermal band was compared with a MODIS LST
product that was acquired on the same day at 10:30 am local time. For this, the Landsat LST
image was resampled to MODIS resolution to enable a pixel to pixel comparison, followed by
extracting the average LST of 7 land cover types using the data set containing the large
delineated plots (Fig. 1).

302        • *Evapotranspiration (ET)*


Based on the Surface Energy Balance Algorithm for Land (SEBAL) (Bastiaanssen, 2000;
Bastiaanssen et al., 1998a, 1998b) we estimated ET (mm/hr) from latent heat fluxes (LE, W/m$^2$)
which were computed as the residual from sensible (H, W/m$^2$) and ground (G, W/m$^2$) heat
fluxes subtracted from net radiation (Rn, W/m$^2$) as:

LE = Rn – G – H                                                                           (4)

We calculated Rn as the sum of incoming shortwave and longwave radiation, minus the
reflected shortwave and longwave radiation and the emitted longwave radiation (equation 5).
The surface albedo, surface emissivity and surface temperature determine the amounts of
incoming and reflected radiation:



$Rn = (1 - \alpha)\, S_d{\downarrow} + \varepsilon_a\sigma T_a{}^4 - (1 - \varepsilon_0)\varepsilon_a\sigma T_a{}^4 - \varepsilon_0\sigma LST^4$           (5)

Where $S_d{\downarrow}$ is the incoming shortwave solar radiation ($W/m^2$) at the surface; $\alpha$ is the surface
albedo (equation 2); $\varepsilon_0$ is the surface emissivity (-); $\varepsilon_a$ is the atmospheric emissivity (-); $\sigma$ is the
Stephan-Boltzmann constant ($5.67 \times 10^{-8}$ $W/m^2/K^4$); LST is the surface temperature (K,
equation 3); $T_a$ is the near surface air temperature (K). The surface emissivity ($\varepsilon_0$) is derived
from the NDVI and is described in the supporting information (Table S1). The average
atmospheric emissivity ($\varepsilon_a$) is estimated with the model of Idso and Jackson, (1969):

$\varepsilon_a = 1 - 0.26 \cdot \exp\{(-7.77 \times 10^{-4}) \cdot (273.15 - T_a)^2\}$           (6)

Ground heat fluxes (G, $W/m^2$) were derived as a fraction of Rn from an empirical relationship
between LST, $\alpha$, and NDVI (Bastiaanssen, 2000) as:

$G = Rn \cdot \dfrac{LST - 273.15}{\alpha} \cdot (0.0038\alpha + 0.0074\alpha^2) \cdot (1 - 0.98NDVI^4)$           (7)

In SEBAL Sensible heat flux (H, $W/m^2$) was calculated as:

$H = \rho Cp \dfrac{\Delta T}{r_{ah}} = \rho Cp \dfrac{a\,LST + b}{r_{ah}}$           (8)

Where $\rho$ is the air density (1.16 $kg/m^3$); Cp is the specific heat of air at constant pressure (1004
J/kg/K); $r_{ah}$ is the aerodynamic resistance to heat transport (s $m^{-1}$); $a$ and $b$ are regression
coefficients which are determined by a hot extreme pixel (where LE = 0 and H is maximum)
and a cold extreme pixel (where H = 0 and LE is maximum). The aerodynamic resistance to
heat transport, $r_{ah}$, is calculated through an iterative process with air temperature measured at 2





m as input. SEBAL is described in Bastiaanssen (2000) and Bastiaanssen et al. (1998a, 1998b).
The application of SEBAL in this research is briefly described in the supporting information
(S3: ET from satellite images).

**2.5 Local short term differences between different land cover types**

From the created LST, NDVI, Albedo and ET images we extracted the average values of the
different land cover classes. For this we used the dataset containing the small 49 delineated
plots covering 7 different land cover types (Fig. 1). The average effect of land transformation,
i.e. the change from forest to another non-forest land cover type, on the surface temperature
was evaluated as (cf. Li et al. (2015)) :

$\Delta LST = LST_{non\text{-}forest} - LST_{forest}$            (1)

A negative $\Delta LST$ indicates a cooling effect and positive $\Delta LST$ indicates a warming effect of
the non-forest vegetation compared to forest. The same procedure was applied in evaluating the
effect of land transformation on the NDVI, albedo and ET.

**2.6 Effects of land cover change on the provincial surface temperature in the past decades**

To analyse the long term effects on the provincial scale we used the MODIS daily LST time
series (MOD11A1 and MYD11A1) from 2000 – 2015. MOD11A1 provides LST for two times
of the day: 10:30 am and 10:30 pm and we used the times series between 2000 and 2015.
MYD11A1 provides LST for 1:30 am and 1:30 pm and is available from 8 July 2002; we used
complete years in our analysis and therefore used the MYD11A1 time series from 2003 – 2015.
We calculated the mean annual LST at four different times of the day (10:30 am, 1:30 pm,



10:30 pm and 1:30 am) between 2000 and 2015 for the lowland of the Jambi from the MODIS
daily LST time series (MOD11A1 and MYD11A1). To do so (1) we calculated for each pixel
the average LST pixel value using only the best quality pixels for every year; (2) from these
pixels we made a composite image (n = 16, one for each year) for the province and (3) from
each composite image we calculated the mean annual lowland provincial temperature as the
average of all the pixels that are enclosed by a zone delineating the lowland of the Jambi
province. We performed the same analysis with the MODIS 16-day NDVI product (2000 –
2015) and the ERA daily temperature grid (2000 – 2015) to compare the annual trends of LST,
NDVI and air temperature of the province. The average provincial LST and NDVI were
compared to the mean LST and NDVI of a selected forest that remained undisturbed forest
during the 2000 – 2015 period.

**2.7 Statistical analysis**

For comparison of the Landsat derived LST and the MODIS LST we analyzed the statistical
relationships with the coefficient of determination ($R^2$), the root mean square error (RMSE),
the mean absolute error (MAE) and the bias (Bias):
$$\text{RMSE} = \sqrt{\frac{\sum_{i=1}^{N}(E_i - O_i)^2}{N}} \qquad (9)$$

$$\text{Bias} = \frac{\sum_{i=1}^{N}(E_i - O_i)}{N} \qquad (10)$$

$$\text{MAE} = \frac{\sum_{i=1}^{N}|E_i - O_i|}{N} \qquad (11)$$





Where $O_i$ is MODIS LST, $E_i$ is the Landsat surface temperature, and N is the number of pixels
compared. Model type 2 linear regression was applied for fitting the relation between MODIS
LST and Landsat LST.
We tested the relation between the biophysical variables LST (or L6 and Rc, both as pre- or
intermediate products before obtaining LST), albedo (α), NDVI and ET with correlation
analysis and a multiple linear regression was applied to analyse the effects of the biophysical
variables on the LST. We used the model: LST (or Rc or L6) ~ α + NDVI + ET, and used $R^2$
and standardized β-coefficients to evaluate the strength of the biophysical variables in
predicting the LST.

**3 Results**

**3.1 Landsat LST compared to MODIS LST**

Landsat and MODIS images showed similar spatial patterns of LST(Fig. 2). In both images the
hot areas correspond to the known clear cut areas, urban areas or other sparsely vegetated areas,
the cooler areas correspond to vegetated areas such as forest, plantation forests and mature oil
palm plantations. The coarse resolution scale of MODIS (1000 m for LST) allows a large
regional coverage of the study area but does not allow to retrieve detailed information on small
patches (smaller than 1 $km^2$). On the other hand, Landsat 7 image allows a detailed study of
patches that are small enough (as small as 30 x 30 $m^2$), but is affected by the scan line error
causing data loss at the edges of the image. In both MODIS and Landsat images clouds and
cloud shadows were removed and therefore lead to data gaps in the images.



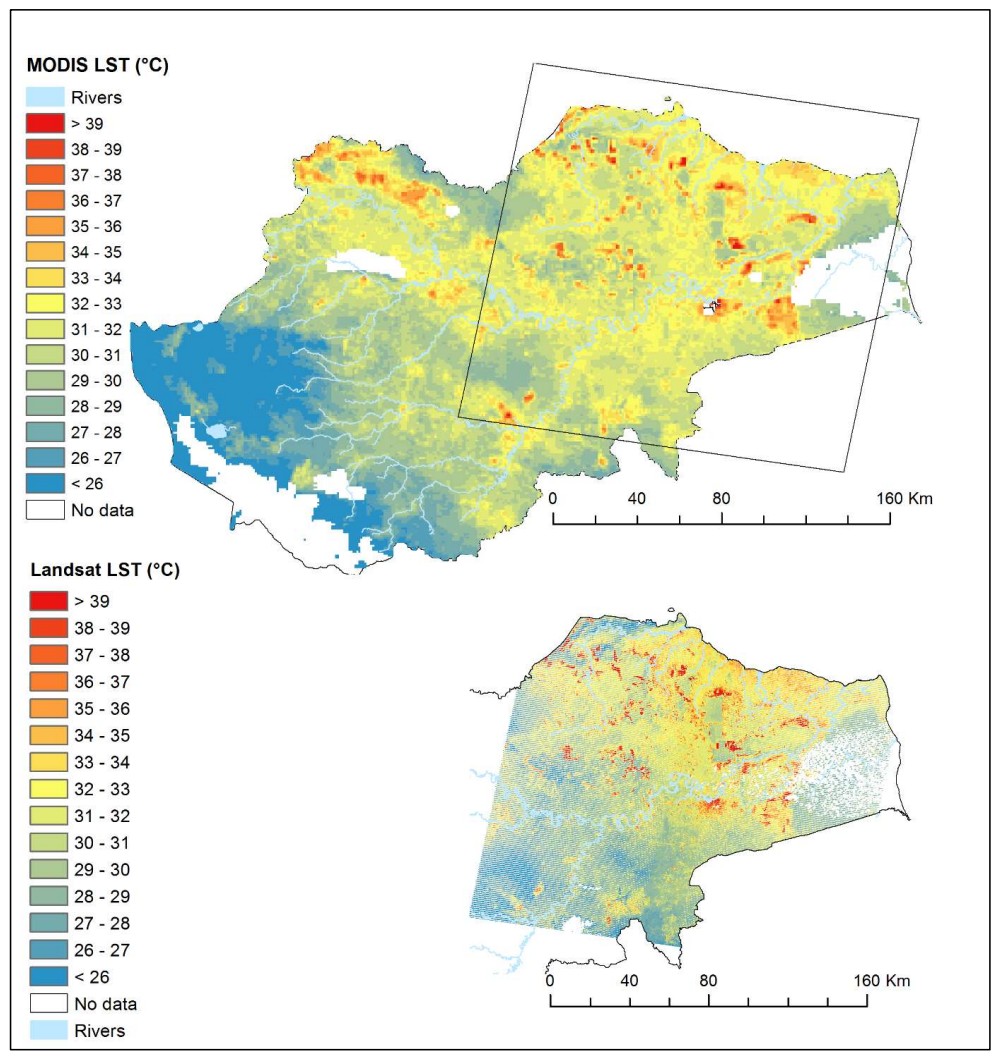

**Fig. 2** MODIS LST image (top) compared with Landsat LST image (bottom). Cloud cover and

cloud shadow cover resulted in data gaps (No data). The difference in acquisition time between

the images is 15 minutes. The square in the MODIS image is the area that is covered by the

Landsat tile (path 125, row 61). Both satellite images were acquired on 19 June 2013.

Landsat derived LST correlated well with MODIS LST ($R^2$ = 0.82; p < 0.001; Fig. 3) with a

RMSE of 1.83 ºC. The 7 land cover types had distinctive LSTs and the observed differences



between these land cover types were consistent in both images. The non-vegetated surfaces
(Clear cut land (CLC) and Urban areas (UB)) had higher surface temperatures than the
vegetated surface types (FO, YOP, MOP, PF and RU). Clear cut land had the highest surface
temperature of all compared land cover types, followed by urban areas whereas the vegetated
land cover types had lower surface temperatures: $LST_{CLC}$ (39.71 ± 2.01 °C ) > $LST_{UB}$ (35.79 ±
1.26 °C) > $LST_{YOP}$ (30.95 ± 0.72 °C) > $LST_{PF}$ (30.25 ± 0.67 °C) > $LST_{MOP}$ (28.98 ± 0.75 °C)
> $LST_{RU}$ (27.78 ± 0.89 °C) > $LST_{FO}$ (27.57 ± 1.41 °C) (Landsat LST, Fig. 3). The same trend
was derived from the MODIS image but with higher surface temperatures, except for CLC:
$LST_{CLC}$ (37.67 ± 1.75 ºC) > $LST_{UB}$ (36.33 ± 1.57 °C) > $LST_{YOP}$ (31.73 ± 0.85 ºC) > $LST_{MOP}$
(30.67 ± 0.88 ºC) > $LST_{PF}$ (29.92 ± 0.93 ºC) > $LST_{RU}$ (29.60 ± 0.36 ºC) > $LST_{FO}$ (29.21 ± 0.40
ºC) (MODIS LST, Fig. 3).

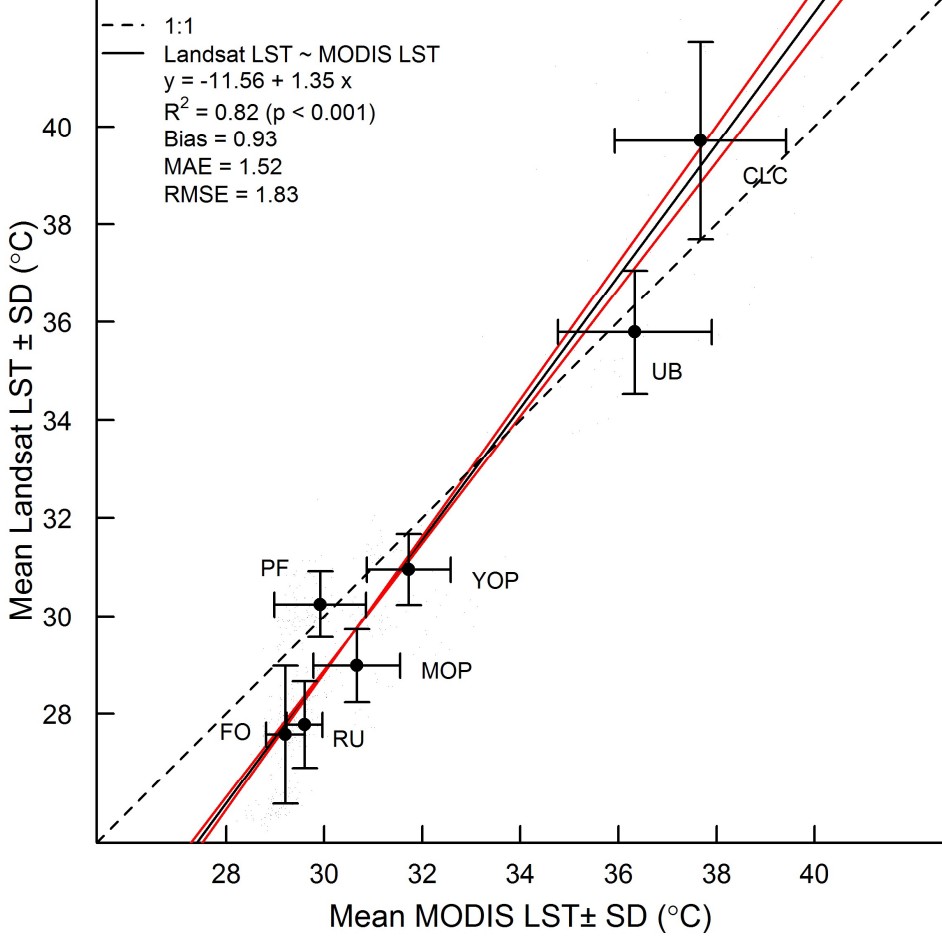


**Fig. 3** Average surface temperature (LST) and standard deviation (SD) of 7 land cover types

derived from Landsat thermal image compared with the mean and SD of MODIS LST.

CLC = Clear cut land, UB = Urban areas, YOP = young oil palm plantation, PF = Acacia

Plantation Forest, MOP = Mature Oil palm plantation, FO = Forest, RU = Rubber plantation.

The dashed line is the theoretical 1:1 line, the solid lines are the Linear Model type 2 regression

line (black) and the confidence limits of the regression line (red). Landsat and MODIS images

were acquired on 19 June 2013, Landsat at 10:13 am local time, MODIS at 10:30 am local time.

Landsat pixels (30 m) were resampled to MODIS pixel resolution (926 m) to make a pixel to





pixel comparison between the two sources possible. RMSE is the root mean squared error, MAE
is mean absolute error.

**3.2 Local short term differences between different land cover types**

The ΔLST between RU, MOP, PF, YOP, UB and CLC land cover types and FO were all
positive, meaning that all other land cover types were warmer than forests (Fig. 4a & Supporting
Information S4 and S5). RU and MOP were $0.4 \pm 1.5$ °C and $0.8 \pm 1.2$ °C warmer than forest,
respectively. PF and YOP were much warmer than forests ($\Delta LST_{PF-FO} = 2.3 \pm 1.1$ °C, $\Delta LST_{YOP}$
$_{-FO} = 6.0 \pm 1.9$ °C). The largest ΔLSTs were between forest and the non-vegetated land cover
types, i.e. UB ($\Delta LST = 8.5 \pm 2.1$ °C) and CLC ($\Delta LST = 10.9 \pm 2.6$ °C). The LST differences
were significant ($p < 0.05$, post-hoc Tukey's HSD test), except between RU and FO ($p = 0.78$,
post-hoc Tukey's HSD test (Supporting Information S6, Table S6.1 & table S6.2).

Similar differences were found for the ΔNDVI between forest and other land covers (Fig. 4b).
The negative ΔNDVI indicates that the non-forest land cover types had lower NDVI than forest.
ΔNDVI between FO and RU, MOP, PF and YOP were small (between $- 0.01 \pm 0.02$
($\Delta NDVI_{MOP-FO}$) and $- 0.12 \pm 0.06$ ($\Delta NDVI_{YOP-FO}$). The largest ΔNDVIs were between forest
and the non-vegetated land cover types, i.e. UB and CLC ($\Delta NDVI = - 0.42 \pm 0.11$ and $- 0.41$
$\pm 0.08$, respectively). All ΔNDVIs were significant ($p < 0.05$, post-hoc Tukey's HSD test).

The difference in albedo (ΔAlbedo) between forest and the other land covers was very small
(Fig. 4c), with ΔAlbedo values between $- 0.03 \pm 0.01$ ($\Delta Albedo_{PF-FO}$) and $0.03 \pm 0.02$
($\Delta Albedo_{YOP-FO}$). These differences were significant ($p < 0.05$, post-hoc Tukey's HSD test).
PF had a lower albedo than forest ($\Delta Albedo_{PF-FO} = - 0.03 \pm 0.01$), while the other land cover
types had a higher albedo than forest.






All land covers had lower ET than forest. RU, MOP and PF had slightly lower ET than FO
($\Delta ET_{RU-FO} = -0.03 \pm 0.04$, $\Delta ET_{MOP-FO} = -0.03 \pm 0.03$ mm/hr, $\Delta ET_{PF-FO} = -0.04 \pm 0.03$ mm/hr)
(Fig. 4d). YOP, UB and CLC had much lower ET values than forests: $\Delta ET_{YOP-FO} = -0.18 \pm$
0.04 mm/hr, $\Delta ET_{UB-FO} = -0.23 \pm 0.04$ mm/hr, $\Delta ET_{CLC-FO} = -0.26 \pm 0.06$ mm/hr). The $\Delta ET$s
were significant ($p < 0.05$, post-hoc Tukey's HSD test).

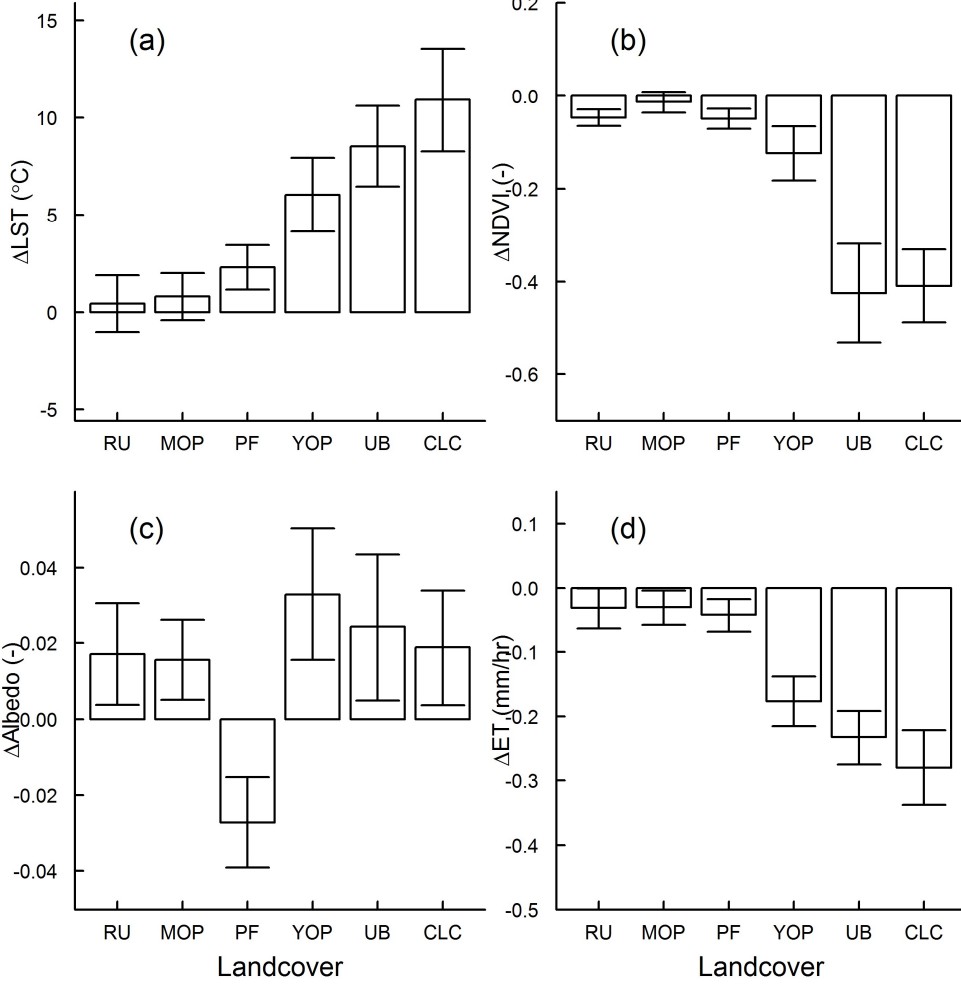

**Fig. 4** Differences (mean ± SD) in surface temperature (ΔLST), normalized difference
vegetation index (ΔNDVI), Albedo (ΔAlbedo) and Evapotranspiration (ΔET) between other



land covers (RU, MOP, PF, YOP, UB and CLC) and forest (FO) in the Jambi province, derived
from the Landsat LST image acquired on 19 June 2013 at 10:13 am local time.

Albedo had the weakest influence on the LST ($\rho = 0.25$, $p < 0.05$) (Table 2) than NDVI and
ET. As the thermal radiance band (L6) and the atmospherically corrected thermal band (Rc)
were the basis for the LST calculation, the high correlation between L6 and NDVI ($\rho = -0.87$,
$p < 0.05$) and between L6 and ET ($\rho = -0.98$, $p < 0.05$) resulted in a high correlation between
LST and NDVI ($\rho = -0.88$) and between LST and ET ($\rho = -0.98$). The analysis showed that
albedo, NDVI and ET were all significant predictors of LST ($F_{(3,\ 41586)} = 1 \times 10^6$, $p < 0.05$). ET
was the strongest predictor of LST (stand. $\beta = -1.11$, $p < 0.05$). Albedo (stand. $\beta = -0.19$, $p <$
0.05, resp.) and NDVI (stand. $\beta = -0.19$, $p < 0.05$) were weaker predictors of LST.

**Table 2** Statistical analysis between biophysical variables (albedo ($\alpha$), NDVI and ET) and
Spectral Radiance band (L6), corrected thermal band (Rc) and Landsat surface temperature
(LST).

| Model | | $\rho$ | $R^2$ | $\beta$ | Stand. $\beta$ | Model fit ($R^2$) | F-statistics |
|---|---|---|---|---|---|---|---|
| **L6 ~ $\alpha$ + NDVI + ET** | $\alpha$ | 0.26 | 0.05 | -2.94 | -0.19 | | F (3, 41586) = |
| | NDVI | -0.87 | 0.10 | 0.23 | 0.11 | 0.99 | $1.10 \times 106$, *** |
| | ET | -0.98 | 1.13 | -4.00 | -1.16 | | |
| **Rc ~ $\alpha$ + NDVI + ET** | $\alpha$ | 0.25 | 0.05 | -4.88 | -0.20 | | F (3, 41586) = |
| | NDVI | -0.88 | 0.04 | 0.16 | 0.05 | 0.99 | $1.79 \times 106$ , *** |
| | ET | -0.98 | 1.00 | -6.21 | -1.10 | | |
| **LST ~ $\alpha$ + NDVI + ET** | $\alpha$ | 0.25 | 0.05 | -34.01 | -0.19 | | F(3, 41586) = |
| | NDVI | -0.88 | 0.05 | 1.30 | 0.05 | 0.99 | $2.3 \times 106$, *** |
| | ET | -0.98 | 1.00 | -43.53 | -1.11 | | |

***: $p = 2 \times 10^{-16}$
LM: Multiple linear regression analysis between LST (or L6 or Rc) and 3 biophysical variables:
Albedo ($\alpha$), NDVI and ET. $\rho$ = correlation coefficient; $R^2$: R-squared of the components; $\beta$ =
regression coefficient of the component; stand. $\beta$ = standardized $\beta$; Model fit ($R^2$): overall model





fit of the multiple linear regression. The values in brackets are for the analysis between the
biophysical variables and the corrected thermal band (Rc).

A separate analysis (Table S6.3, Supporting information S6) showed that ET was a strong
predictor of LST for each land cover type in this study and that NDVI and albedo were minor
predictors of LST.

**3.3 Effects of land-use change on the provincial surface temperature in the past decades**

The average annual LST of the province was characterized by a fluctuating but increasing trend
during daytimes (Fig. 5a and 5b) between 2000 and 2015. The average morning LST (10:30
am) increased by 0.07 ºC per year ($R^2 = 0.59$; $p < 0.0001$), the midday afternoon LST (13:30
local time) increased by 0.13 °C per year ($R^2 = 0.35$; $p = 0.02$) between 2003 and 2015. While
the daytime LST showed a clear increase, the night and evening LST (10:30 pm and 1:30 am,
Fig. 5c and 5d) trends were small showing a decrease of – 0.02 °C ($R^2 = 0.29$; $p = 0.02$) and –
0.01 °C ($R^2 = 0.05$; $p = 0.51$) per year, respectively. The observed LST trends resulted in a total
LST increase of 1.05 °C and 1.56 °C in the morning (10:30 am) and afternoon (1:30 pm)
respectively and a total decrease of the province LST of 0.3 °C (10:30 pm) and 0.12 °C (1:30
am) at night over the time period 2000 to 2015.

In order to separate the effect of land use change from global climate warming, we used a site
constantly covered by forest over that period (from the forest sites we used in this study) as a
reference  not directly affected by land cover changes. That site showed less changes in LST
than the entire province:  only the mean morning LST (10:30 am) had a significant but small
trend with an increase by 0.03 °C per year ($R^2 = 0.21$, $p < 0.05$) resulting in a total LST increase
of the province of 0.45 °C between 2000 and 2015 (Fig. 5a). This LST warming is much smaller



than the overall warming at provincial level of 1.05 °C. The LST time series at other times
showed no significant trends: the mean afternoon LST (1:30 pm) with -0.05 °C per year ($R^2$ =
0.01, p = 0.31) (Fig. 5b), the night and evening LST with 0.01°C per year (Fig. 5c and 5d, p =
0.19 and p = 0.65, respectively).

The mean annual NDVI of the province decreased by 0.002 per year which resulted in a total
NDVI decrease of 0.03 ($R^2$ = 0.34; p = 0.01; Fig. 5e). The NDVI of the forest showed a small
but not significant increase of 0.001 per year ($R^2$ = 0.04, p = 0.23) (Fig. 5e) fluctuating around
an NDVI of 0.84.

The mean annual midday air temperature (at 1:00 pm, local time, Fig. 5f) and the mean annual
night air temperature (at 1:00 am, local time) increased every year by 0.05 °C and 0.03 °C,
respectively resulting in a total air temperature increase of 0.75 °C ($R^2$ = 0.66, p < 0.0001) and
0.45 °C ($R^2$ = 0.32, p = 0.014) between 2000 and 2015 (Fig. 5f).





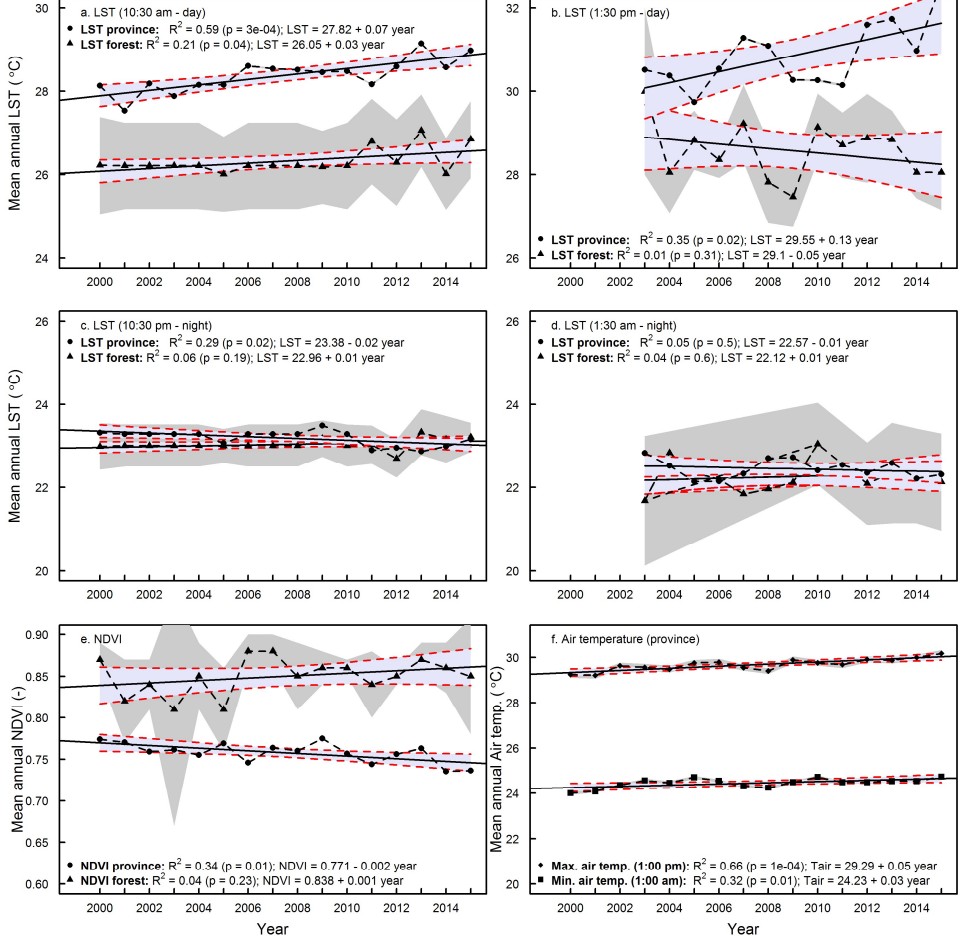


**Fig 5**. Mean annual LST (a – d), mean annual NDVI (e) and mean annual air temperature trends
(f) in the Jambi province between 2000 and 2015 derived from MODIS LST (5a. 10:30 am, 5b.
1:30 pm, 5c. 10:30 pm and 5d. 1:30 am, local time), MODIS NDVI and ERA Interim Daily air
temperature (1:00 am and 1:00 pm, local time) data sets respectively. Grey-shaded areas are the
confidence intervals of the means, blue-shaded areas are the confidence intervals of the
regression lines. MODIS LST time series for 1:30 pm and 1:30 am were available from the mid
of 2002; for this reason we used the complete years from 2003 till 2015.

549

**4 Discussion**



551

## 4.1 Landsat LST compared to MODIS LST

553

In our study we retrieved the surface temperature from a Landsat image and compared this with

MODIS LST. Our results showed a good agreement between both LSTs (Fig. 3), which is

comparable to other studies and thus gives confidence in our analysis. Bindhu et al. (2013)

found also a close relationship between MODIS LST and Landsat LST using the same

aggregation resampling technique as our method and found $R^2$ of 0.90, a slope of 0.90, and an

intercept of 25.8 for LST, compared to our $R^2$ of 0.8, slope of 1.35 and intercept of –11.58 (Fig.

3). Zhang and He (2013) validated Landsat LST with MODIS LST and also found good

agreements (RMSD 0.71 – 1.87 ºC) between the two sensors, where we found a RMSE of 1.71

ºC. Nevertheless, there still are differences and slope versatility between the two satellite

sources. These differences are typically caused by differences between MODIS and Landsat

sensors in terms of (a) different sensor properties e.g. spatial and radiometric resolution and

sensor calibration; (b) geo-referencing and differences in atmospheric corrections (Li et al.,

2004); and (c) emissivity corrections i.e. the use of approximate equations to derive the

emissivity from the NDVI from Landsat's Red and NIR bands. Li et al. (2004) and Vlassova et

al. (2014) identified these same factors in their comparison of ASTER LST with MODIS LST

and Landsat LST with MODIS LST, respectively. Vlassova et al. (2014) found good

agreements between MODIS and Landsat LST with MODIS LST to be higher than Landsat

LST, which they attributed to the delay of 15 minutes in acquisition time between MODIS and

Landsat. MODIS LST is measured 15 minutes later and our results showed that MODIS LSTs

were indeed higher than Landsat LST. A comparison of MODIS LST with locally measured

canopy surface temperatures during the overpass time of MODIS also showed agreement

(Supporting information S7, Figure S7.1). The slope was possibly due to differences in



instrumentation and emissivity corrections and to scale issues, still this comparison could
corroborate the quality check of MODIS LST.
As the MODIS LST product is proven to be accurate within 1 ºC (Silvério et al., 2015; Wan et
al., 2004) and has been intensively validated, the use of MODIS LST was a proper way to assess
the quality of our Landsat LST.

The errors from the different sources (such as atmospheric correction, emissivity correction,
resampling Landsat to MODIS resolution) are difficult to quantify. When we tested the impact
of atmospheric correction and emissivity errors on the LST from Landsat retrieval we found
that: (a) the overall patterns across different land use types did not change, (b) emissivity was
the most important factor but the effects on LST retrieval were small and (c) errors due to
atmospheric correction parameters were small because there were small differences between
default Atmospheric correction (ATCOR) parameters and ATCOR parameters derived with
actual local conditions (relative humidity (RH), air pressure and air temperature). Following
the method of Coll et al. (2009) and Jiang et al. (2015) we show that the use of the online
atmospheric correction parameter calculator is a good option provided that RH, air temperature
and air pressure are available. We additionally compared locally measured air temperatures
with MODIS air temperature and found a good agreement (Supporting information S8, Figure
S8.1), which served as a verification that we used a correct air temperature for the atmospheric
correction parameter calculator.
Overall, our comparison of LST from Landsat against LST from MODIS as well as against
ground observation suggests that we are able to retrieve meaningful spatial and temporal
patterns of LST in Jambi province.

**4.2 LST patterns across different LULC types**



The land cover types in our study covered a range of land surface types that develop after forest
conversion. This is the first study in this region that includes oil palm and rubber as land use
types that develop after forest conversion. The coolest temperatures were at the vegetated land
cover types while the warmest surface temperatures were on the non-vegetated surface types
like urban areas and bare land. Interestingly, the oil palm and rubber plantations were only
slightly warmer than the forests whereas the  young oil palm plantations had clearly higher LST
than the other vegetated surfaces. For other parts of the world, Lim et al. (2005, 2008), Fall et
al. (2010) and Weng et al. (2004) also observed cooler temperatures for forests and the highest
surface temperatures for barren and urban areas.
In Indonesia, land transformation is often not instantaneous from forest to oil palm or rubber
plantation, but can be associated with several years of bare or abandoned land in-between (Sheil
et al., 2009). Oil palm plantation typically have a rotation cycle of 25 years, resulting in
repeating patterns with young plantations (Dislich et al., 2016). Given the large differences in
LST between forests and bare soils or young oil palm plantations that we observed, a substantial
warming effect of land transformation at regional scale is expected.


**4.3 Drivers of local differences between different land cover types**

All land cover types (except Acacia Plantation Forests) had a higher albedo than forest,
indicating that these land cover types absorbed less incoming solar radiation than forests.
Nevertheless, these land cover types were warmer than forests, suggesting that the albedo was
not the dominant variable explaining LST. Indeed, the statistical analysis showed that ET ~
LST had a higher correlation than albedo ~ LST. The ΔETs were significant, underlying that
despite their higher albedo all land cover types had higher LSTs than forests due to lower ET
rates than forests. Vice versa, forests that absorb more solar radiation due to the lower albedo,

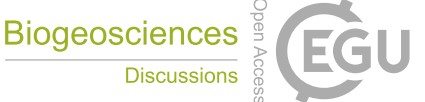

have lower LST due to the higher ET they exhibit, hereby identifying evaporative cooling as
the main determinant of regulating the surface temperature of all vegetation cover types (Li et
al., 2015).

Both observational and modeling studies carried out in other geographic regions and with other
trajectories support our observations. Observational studies in the Amazonia by Lawrence and
Vandecar (2015) on the conversion of natural vegetation to crop or pasture land showed a
surface warming effect. Salazar et al. (2015) provided additional evidence that conversion of
forest to other types of land use in the Amazonia cause significant reductions in precipitation
and increases in surface temperatures.
Alkama and Cescatti (2016) and earlier studies by Loarie et al. (2011a, 2011b) showed that
tropical deforestation may increase LST, croplands in the Amazonian regions were also warmer
than forests through the reduction of ET (Ban-Weiss et al., 2011; Feddema et al., 2005) and that
the climatic response strongly depends on changes in energy fluxes rather than on albedo
changes (Loarie et al., 2011a, 2011b). A study by Silvério et al. (2015) indeed found that
tropical deforestation changes the surface energy balance and water cycle and that the
magnitude of the change strongly depends on the land uses that follow deforestation. They
found the LST over croplands 6.4 ºC higher and over pasture lands 4.3 ºC higher compared to
the forest they replaced, caused by energy balance shifts. Ban-Weiss et al. (2011) and Davin
and de Noblet-Ducoudré (2010) added that in addition to the reduction of ET, the reduction of
surface roughness most likely enhanced the substantial local warming.

Also for non-Amazonian regions the replacement of forests by crops resulted in changes similar
to our observations. In temperate Argentina, Houspanossian et al. (2013) found that the
replacement of dry forests by crops resulted in an increase of albedo and still the forests
exhibited cooler canopies than croplands. The cooler canopies were a result of the higher



aerodynamic conductance that caused the capacity of tree canopies to dissipate heat into the
atmosphere and that both latent and sensible heat fluxes operate simultaneously cooling forest
canopies Houspanossian et al. (2013).

In a global analysis Li et al. (2015) showed that tropical forests generally have a low albedo,
but still the net energy gain caused by solar energy absorption is offset by a greater latent heat
loss via higher ET and that in the tropical forests the high ET cooling completely offsets the
albedo warming. For China, this cooling effect was also shown by Peng et al. (2014) who
compared LST, albedo and ET of plantation forests, grassland and cropland with forests.

For the USA,  Weng et al. (2004) and for China, Yue et al. (2007)  used NDVI as a vegetation
abundance indicator and also found areas with a high mean NDVI to have lower LST than areas
with a low mean NDVI, all suggesting that vegetation abundance is an important factor in
controlling the LST through higher ET rates. Our result support their assumptions by showing
the high correlation between NDVI – LST and ET – LST.

Our findings are also supported by modelling studies. Beltrán-Przekurat et al. (2012) found for
the Southern Amazon that conversion of wooded vegetation to soy bean plantations caused an
increase of the LST due to decreased latent heat and increased sensible heat fluxes. Climate
models also show the same warming trends and land surface modelling also project an increase
in surface temperatures following deforestation in the Brazilian Cerrado (Beltrán-Przekurat et
al., 2012; Loarie et al., 2011b). In a global analysis, Pongratz et al. (2006) showed the LST
increase of forest to cropland or pasture transitions, also driven by reduced roughness length,
increased aerodynamic resistance, and that the temperature response is intensified in forest to
clear land or bare land transitions (1.2 ºC increase). Similar to observational studies, the



modelling results of Bathiany et al. (2010) show that ET is the main driver of temperature
changes in tropical land areas.

In understanding the effects of deforestation on biophysical variables in Indonesia, our study
identifies the following mechanisms: (a) reduction of ET decreases surface cooling, (b) reduced
surface roughness reduces air mixing in the surface layer and thus vertical heat fluxes, (c)
changes in albedo change the net radiation, (d) changes in energy partitioning in sensible and
latent heat and heat storage. The effect is an increase of the mean temperatures leading to
warming effects in all tropical climatic zones (Alkama and Cescatti, 2016). We point here that
our study (1) included a ground heat flux, but did not take into account the storage of heat in
the soil and the release of stored heat out of the soil during the daily cycle and (2) that the
Landsat satellite image was obtained under cloud free conditions with high shortwave radiation
input and low fraction of diffuse radiation. Therefore, the LST retrieved on cloud free days
might be overestimated compared cloudy days where the differences in LST between land uses
are supposed to be less when diffuse radiation increases.
Our study is the first to include the oil palm and rubber expansion in Indonesia. In Indonesia,
smallholders take 40% of the land under oil palm cultivation for their account (Dislich et al.,
2016). Since the landscape in the Jambi province is characterized by small-scale smallholder-
dominated mosaic including rubber and oil palm monocultures (Clough et al., 2016), studies
using medium to coarse resolution data are not able to capture the small scale changes and
processes at the small-scale level. By using high resolution Landsat data we were able to also
include the effects of land use change on biophysical variables and the underlying processes of
the small scale holder agriculture.



**4.4 Effects of land use change on the provincial surface temperature in the past decades**



The mean surface temperature of the Jambi province increased stronger during the morning
(10:30 am) and afternoon (1:30 pm) than during the evening (10:30 pm) and night (1:30 am).
Given that our results show a decrease of the NDVI in the same period, this suggests that the
observed increased trend of the day time province LST can be attributed to land cover changes
that occurred. Our assumption that the observed decreasing NDVI trend is caused by land
conversions is supported by two different studies which reported that in the Jambi province
between 2000 and 2011 (Drescher et al., 2016) and between 2000 and 2013 (Clough et al.,
2016) the forest area decreased and that the largest increases were for rubber, oil palm, and
agricultural and tree crop areas. The class 'other land use types' which includes urban areas
showed a minor increase (around 1%) which suggests that the decrease in NDVI was most
likely caused by forest cover loss and not by urban expansion (see Supporting information,
Table S9). The same observations on LULC change in Indonesia were also supported by Lee
et al. (2011), Margono et al. (2012, 2014), Paterson et al. (2015) and Luskin et al. (2014). Luskin
et al. (2014) showed that in the period 2000 – 2010 forests decreased by 17%, oil palm and
rubber area increased by 85% and 19%, respectively, in the Jambi province.

Given these trends in LULC changes, the observed LST trends were most likely caused by
gradual decrease of forest cover loss at the expense of agriculture and croplands. Our
assumptions are supported by findings of Silvério et al. (2015), Costa et al. (2007), Oliveira et
al. (2013), Spracklen et al. (2012) and Salazar et al. (2015) which indicate that land use
transitions in deforested areas likely have a strong influence on regional climate. Alkama and
Cescatti's (2016) analysis show that biophysical effects of changes in forest cover can
substantially affect the local climate by altering the average temperature, which is consistent
with our observations and can be related to the observed land use change in the Jambi province.



As Indonesia has undergone high rates of forest cover loss from 2000 to 2012 (Margono et al.,
2014), these findings support our assumptions that the observed LST increase in the Jambi
province was most likely caused by the observed land use changes.

To separate the effect of global warming from land-use change induced warming, we
considered areas with permanent and large enough forests as reference where changes are
mainly due to global warming. We find that LST of forests show either no significant trends (at
1:30 pm, 10:30 pm, 1:30 am) or just a clearly smaller increase of 0.03 °C per year at 10:30 am.
The difference between the LST trend of the province and of the forest at 10:30 am was 0.04
°C per year, resulting in a ΔLST of 0.6 °C between the province and forest in the period 2000
and 2015. Using the warming effects we found between forest and other land cover types
(ΔLST, Fig. 4a) and the observed land cover changes by Clough et al. (2016), Drescher et al.
(2016) (Supporting Information S9, table S9.1 and S9.2) we estimated the contribution of all
land cover types (except forest) to the ΔLST of the province between 2000 and 2015 to be
0.51°C out of 0.6°C observed above, which also supports our assumption that the increase of
the province LST was by 85% driven  by land cover changes (see Supporting Information 9,
Table S9.1 & S9.2: Land use change analysis), with clear cut areas having a large contribution
as they have the largest warming effect.

The observed small, but significant increase in LST of forests by 0.03 °C per year at 10:30 am
reflects a LST change independent to land cover changes as the forest remained unchanged over
that time period. Potential driver of that LST increase is the general global air temperature trend
due to changes in radiative forcing or border effects (advection from warmer land uses), which
is  similar to the 1994 - 2014 time series analysis of Kayet et al. (2016) – who showed  a LST
increase for all land cover types ranging from wasted land, agriculture land, open forest, dense
forest, water bodies, built up.




The observed trends of province air temperature (Fig. 5f) were significant, suggesting that a
general warming due to global and regional effects contributes to the observed warming at
province level during day and night time, but is smaller than the land cover change induced
effects (Supporting Information S9, Table S9.1 & S9.2) at provincial level (Fig. 5a and 5b).

In our long term analysis on the regional effects of land use change we observed an increase in
the mean LST and mean air temperature in the 2000 - 2015 period, concurrent to a decrease of
the NDVI. The warming observed from MODIS LST data and from the air temperature
obtained from the independent ERA Interim Reanalysis in the Jambi province are most likely
caused by the observed decrease of the forest area and an increase oil palm, rubber and other
cash crop areas in the same period, with other effects such as radiative forcing changes and
additional natural effects playing a smaller role. Given the plan of the Indonesian governmental
to substantially expand oil palm productivity with an projected additional demand of 1 to 28
Mha in 2020 (Wicke et al., 2011), the strong warming effect we show for Jambi province may
serve as an indication of future changes in LST for other regions of Indonesia that will undergo
land transformations towards oil palm plantations.
The observed effects of land use change on the biophysical variables may have implications for
ecosystem services in the Jambi province beyond a pure warming effect. The high precipitation
in this region in combination with the reduced vegetation cover of bare land and young oil palm
plantations impose risks of soil erosion caused by surface run off. Less water infiltrates in the
soil, thereby decreasing the soil water storage that may lead to low water availability in the dry
season (Dislich et al., 2016; Merten et al., 2016). High surface temperatures in combination
with low water availability may make the vegetation and the surroundings more vulnerable for
fires.





**5 Conclusion**

In summary, we showed the importance of forests in regulating the local and regional climate. We derived biophysical variables from satellite data, analyzed the biophysical impacts of deforestation and on a local scale we found a general warming effect after forests are transformed to cash or tree croplands (oil palm, rubber, acacia) in the Jambi province of Sumatra. The warming effect after forest conversion results from the reduced evaporative cooling, which was identified as the main determinant of regulating the surface temperature. On a regional scale, we saw that the effects of land cover changes are reflected back in changes of the LST, NDVI and air temperature of the Jambi province. The warming effect induced by land cover change clearly exceeded the global warming effect. Understanding the effects of land cover change on the biophysical variables may support policies regarding conservation of the existing forests, planning and expansion of the oil palm plantations and possible afforestation measures.



*Author contributions*. Clifton R. Sabajo conducted the research, fieldwork an analysis and
prepared the manuscript, which was reviewed by Guerric le Maire, Tania June, Ana Meijide,
Olivier Roupsard and Alexander Knohl. Ana Meijide and Alexander Knohl provided the
meteorological data.

      *Competing interests*. The authors declare that they have no conflict of interest.

*Acknowledgements*. This research was funded by the Erasmus Mundus Joint Doctorate
Programme Forest and Nature for Society (EMJD FONASO) and the German Research
Foundation (DFG) through the CRC 990 "EFForTS, Ecological and Socioeconomic Functions
of Tropical Lowland Rainforest Transformation Systems (Sumatra, Indonesia)" (subproject
A03). A special thanks to Huta Julu Bagus Putra, a.k.a. Monang, for his assistance and
translation during the field work in Indonesia. The authors state to have no conflict of interest.






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
