# Peer review of "Expansion of oil palm and other cash crops causes an increase of the land surface"

_Biogeosciences, 2017_

## Short Comment (SC1) · 31 May 2017

I am unclear how the data in Paterson et al. (2015) supported LULC change in Indonesia as was stated in the manuscript. The authors should clarify. They could also write LULC in full at first with the abbreviation in brackets afterwards.

The English could be tightened up in parts of the paper. E.g. they start sentences with "We" on lines 31 and 34. Also, this is not the passive voice. Similarly, "Our" lines 40 and 43. Line 57: "in the past decades" is imprecise and clumsy. Lines 66-69: Three "and"s are used in this sentence. Not the best English. Line 75: Replace "rise" with "increase". Line 94-96: You should not have to explain what the "vice versa" is. Anyway, I will stop there as its too much work to point out other such smallish things.

[Figure]

Maybe you consider these unimportant...

---

## Short Comment (SC2) · 21 Jun 2017

Given the nature, impact and extend of results presented the paper submitted by Sabajo et al, the present title of article clearly appears as unapropriate. Indeed results do not show any pivotal role for oil palm cultivation which should support the assumption that it is a key driver of phenomena observed. The title should stick only to facts and findings, which tend to evidence a difference between native forest and cultivated land in terms of land surface temperature.

---

## Referee Comment (RC1) · Anonymous Referee #1 · 3 Jul 2017

Review: Expansion of oil palm and other cash crops causes an increase of land surface temperature in Indonesia

General: The authors investigate the effect of land cover change (from forests to 'other' and mostly oil palm plantations) on regional land surface temperatures. They use remote sensing to determine LST (and albedo, NDVI, and ET). They conclude that conversion from forests had led to a ∼1 degree C temperature (positive) change after accounting for albedo. They also conclude that this is a positive feedback to climate warming. I only suggest some minor edits and (if the authors can) and expansion of the discussion of what these LST changes might translate to in the atmosphere? How

much larger of a region will they affect? How would you determine this?

Introduction: nicely written and I appreciate the well thought out definitions. Line 96: missing an "as" after "such" Methods: Could you describe the study sites in a little more detail (rather than the reference Drescher) 1. ET calculations: I'm familiar with the use of satellite data for all of the variables except for ET. Did you compare ET with the tower sites? How well does it work? I see that you added this to the supplement, but it would be nice to have a validation of this method explained in the main text.

Results: Line 405-406: Hot = red? And cool = Blue colors. Can you please specify this?

Discussion: Line 668: When I look at the figures, there also seems to be a high correlation between NDVI and ET (simply because the response pattern, the pattern of the changes, look very similar). Can you explain this? Is it because of the ET calculation? Line 763: "concurrent to" should be "concurrent with" Line 768: "governmental" should be "government"

Final remarks: This is a well-written, well-organized manuscript. I support publication in biogeosciences.

———————————————————

---

## Referee Comment (RC2) · Anonymous Referee #2 · 16 Jul 2017

General comments: Sabajo et al. evaluates the impact of land use changes on land surface temperatures in Indonesia over the MODIS timespan (1999-2015). The study is well written and provides a good, long-term observational analysis clearly showing the impact of regional deforestation on increasing land surface temperature across an entire region. The only general comment I have is that it would be good to include a seasonality analysis showing how deforestation has changed land surface temperature in both wet and dry season. I know that satellite remote sensing is more challenging during the wet season, but I think evaluating the impact of land changes with season-ality would be useful. This could also highlight likely reductions in ET with land change (and shallower rooting zones) during the dry season. The dry season is also when

heat impacts (including wildfires) could be more significant.

Specific comments: For Figure 1 (and text in the manuscript related to Figure 1), it might be good to describe the general atmospheric circulation for wet and dry seasons (are winds from the east or from the west). This would help the reader evaluate whether there are substantial land use changes upwind of the forest plots that are used as the baseline "control" to evaluate land surface temperature changes to due land use changes and not overall global climate change.

---

## Author Comment (AC4) · 22 Aug 2017

We agree that a seasonality analysis might show differences between the wet and dry season. We now made a seasonality analysis. Overall, the relationships in the dry season are stronger than for the wet season as we have much more usable data during the dry season. Only for the 10:30 am data we have significant relationships for both the dry as well as wet season. LST increased 0.08 $\pm$ 0.01 degrees C per year during the dry season, while the increase during the wet season was lower (0.05 $\pm$ 0.01 degrees C per year) (fig. 1). This suggests that the warming is more pronounced during the dry season compared to the wet season, which is reasonable as we have more

incoming radiation during the dry season. Nevertheless, we prefer to pool the data from the dry and the wet season in order to get more statistically robust relationships.

We added the following sentence to the manuscript (line 755): "We like to point out that our MODIS analysis has a larger proportion of data from the dry season compared from the wet season, as there were more cloud free conditions during the dry season. Thus, our reported warming effect reflects cloud free conditions. During cloudy conditions, particularly in the wet season, the warming effect is expected to be lower."

In our analysis of the MODIS LST data we have not come across anomalous LST that could be attributed to forest fires. This is caused by the mask we applied in selecting the best quality pixels which mostly also removed pixels covered by smoke. A seasonality analysis is not possible with Landsat data because there is not enough data.

We include a wind rose (Fig. 2) from one of our reference meteorological stations in the area, (see Drescher et al., 2016), for data collected between October 2013 to May 2016. Based on the climate diagram for the region (obtained from data from 1991-2011) we considered as the dry season the months of June-September and the rest was considered as wet season. See the wind roses for the dry (left) and wet (right) seasons below:

During the dry season winds were predominantly from the SE, whereas during the wet season winds where predominantly from the NW. The SE vs. NW shift in wind directions is in line with the regional monsoonal circulation. The landscape in the lowland of Jambi province is, however, very patchy with small-scale mosaics of different land uses. While we cannot fully rule out that advection from upwind land use changes may play a role, but it seems unlikely to have a systematic bias given the typical patchiness of the landscape. Also warm air advection would mean that the "climate change" warming of the forested "control" site is overestimated, thus making the land-use change effect even larger.

We thank the anonymous referee for reviewing the manuscript and for the suggestions

to improve the manuscript.

[Figure]

[Figure]

[Figure]

**Fig. 1.** Mean annual LST (a – d) separated in wet and dry season.

[Figure]

**Fig. 2.** Wind rose from a reference meteorological stations in the study area (Jambi province).

---

## Author Response (AR2)

**Associate Editor Decision: Publish subject to minor revisions (Editor review)** (14 Sep 2017) by Paul Stoy

Comments to the Author:
Responses to referees were comprehensive and thank you for the attention to detail. There are a few areas where the manuscript would benefit from copy-editing or minor editorial improvements that would improve accuracy and readability.
*We thank the associate editor for reviewing the manuscript and for pointing out the errors in some details we overlooked. We are happy to apply the suggestions for improving the manuscript.*

The following is a partial list of minor revisions that could be made to improve the manuscript.

1. I recommend a couple more careful reads by the authors before recommending the manuscript for publication.
*We have gone through the manuscript and filtered out many mistakes we overlooked in the previous versions of the manuscript.*

2. For the title I might recommend "in Jambi province"
*We have adopted the suggested change. The title is now:*
*Expansion of oil palm and other cash crops causes an increase of the land surface temperature in the Jambi province in Indonesia.*

3. "Surface roughness affect the transferred sensible and latent heat" on line 88 is an example of verb/subject agreement that needs to be changed.
*We thank the associate editor for pointing us to this error. We corrected the sentence.*

4. The statement on line 92 would benefit from a reference.
*We added a reference to underline the statement in this line and rewrote the sentence to ".. LST is a major land surface parameter and as climatic factor it is regarded a main driver of diversity gradients related to the positive relationships between temperature and species richness (Wang et al., 2016)."*

*Wang, J., Pan, F., Soininen, J., Heino, J. and Shen, J.: Nutrient enrichment modifies temperature-biodiversity relationships in large-scale field experiments, Nat. Commun., 7, 13960, doi:10.1038/ncomms13960, 2016.*

5. On line 97, this is true "all else being equal" because soil and ecosystem heat flux and net radiation of course changes as well with a change in vegetation.
*We rewrote this paragraph completely and added the condition as suggested in the new paragraph. See also #6.*

6. The paragraph beginning line 105 is a bit vague and out of place. It could be instead integrated into the previous paragraphs to provide context.
*We rewrote this paragraph completely and integrated it with the previous paragraph. See also #5. The restructured paragraph is:*

*"The replacement of natural vegetation also changes evapotranspiration (ET) (Boisier et al., 2014) and LST because the surface biophysical variables (i.e. surface albedo, LST, emissivity and indirectly Leaf Area Index (LAI) and Normalized Difference Vegetation Index (NDVI)) are interconnected through the surface radiation balance. When ET decreases for example, surface temperatures and sensible heat (H) fluxes increase; on the other hand, when ET increases, the increased LE fluxes lower surface temperatures and decrease H fluxes (Mahmood et al., 2014) under equal net radiation conditions because with a change in vegetation, soil and ecosystem heat flux and net radiation also change due to an alteration of the biophysical variables. Vegetation structure, represented by NDVI, LAI and vegetation height, is in this respect an important determinant of the resistances or conductivities to heat, moisture, and momentum transfer between the canopy and the atmosphere (Bright et al., 2015) facilitating the amounts/ratios of sensible heat to water vapour dissipation away from the surface (Hoffmann and Jackson, 2000)."*

7. Equation 2 should have subscripts.
8. On 302 the degree symbol is not necessary before K
*We thank the associate editor for pointing to these details. We added the subscripts (comment #7) and removed the degree symbol before K (comment #8).*

9. Equation 5 would be more accurate as sky temperature rather than near surface air temperature. That being said in this case the difference wouldn't be very large.
*We changed the term to 'sky temperature'. The term sky temperature is more appropriate because we used the temperature measured 2 m above the ground.*

10. Please use the multiplication sign rather than the dot (which can mean dot product) in equation 6 and elsewhere and please also use standard parentheses rather than curly braces.
*We thank the associate editor for pointing to this detail. We changed the equations as suggested.*

11. 415: relatively hot and cool.
*We changed the description of the images as suggested.*

12. I find 4 significant digits for the values starting line 440 to be a bit too accurate given that these are from remote sensing platforms. 3 significant digits is better but still probably a bit generous.
*We applied this suggestion and use now 3 significant digits. We applied this to other parts in the manuscript too.*

13. capital 'Atmosphere' on line 603.
*We corrected this.*

14. 816 could be more specific.
*We removed the general statement of this first sentence of the conclusion, rewrote it added specific information related to the study area. We rewrote the first sentence of the conclusion as:*

*"In summary, we studied the effects of land use and land cover changes on the surface biophysical variables in Jambi and explained the underlying mechanisms of the surface temperature regulation."*

**I. R. Paterson (russell.paterson@deb.uminho.pt)**

1. I am unclear how the data in Paterson et al. (2015) supported LULC change in Indonesia as was stated in the manuscript. The authors should clarify.
*Line 718 (in the discussion): Paterson et al. (2015) mention that the LST in Malaysia increased over the last four decades of 2.7 – 4.0 degrees C per 100 years. This is an example of LST change in another region close to Indonesia. However, this is not from first hand observation, data or research but results from the Malaysian Meteorological Department. We admit that this reference is misplaced here and removed the reference to Paterson et al. (2015) from the discussion.*

2. They could also write LULC in full at first with the abbreviation in brackets afterwards.
*We realize that the abbreviation LULC is introduced in line 600 and that it was not defined anywhere else. We added the full words followed by the abbreviation in line 600 as "LST patterns across different land use and land cover (LULC) types.*

3. The English could be tightened up in parts of the paper.
a. E.g. they start sentences with "We" on lines 31 and 34. Also, this is not the passive voice. Similarly, "Our" lines 40 and 43.
*We have tried to improve the English in the manuscript, removing the repeated words in the same paragraph, such as the "we" or "our" in the indicated lines. We have especially tightened the abstract, but kept it in active voice, as the Biogeosciences' guidelines do not explicitly mention the preferred style of writing.*

b. Line 57: "in the past decades" is imprecise and clumsy.
*We changed this to "in the past two and a half decades" to refer to our study period 2000 - 2015*

c. Lines 66-69: Three "and"s are used in this sentence. Not the best English.
*The biophysical variables (albedo, emissivity and surface roughness) affect gas and energy exchange processes between the land surface and the atmosphere. We did not find any obsolete "and" in this construction.*

d. Line 75: Replace "rise" with "increase".
*We changed the verb as suggested.*

e. Line 94-96: You should not have to explain what the "vice versa" is.
*We removed the "vice versa" and connected the remaining sentence to the previous. We wanted to describe both situations of ET increase and ET decrease: what happens when ET increases and when ET decreases. The rephrased sentence is now: "In case ET is decreased, surface temperatures and fluxes of sensible heat (H) increase. On the other hand, when ET increases, increased LE fluxes lower surface temperatures and decrease H fluxes (Mahmood et al., 2014)."*

f. Anyway, I will stop there as its too much work to point out other such smallish things. Maybe you consider these unimportant...
*We thank Dr. Paterson for his comments and suggestions for improvements. We are happy to incorporate the suggested language corrections. Additionally, we reread the entire manuscript and made further English correction.*

**II. A. RIVAL (alain.rival@cirad.fr)**

1. Given the nature, impact and extend of results presented the paper submitted by Sabajo et al, the present title of article clearly appears as unapropriate. Indeed results do not show any pivotal role for oil palm cultivation which should support the assumption that it is a key driver of phenomena observed. The title should stick only to facts and findings, which tend to evidence a difference between native forest and cultivated land in terms of land surface temperature.

*We agree with Dr. Rival that the title is too general. Our study shows results from Jambi province/Indonesia only, thus we now mention this specifically in the title. We however think that the evidence from our study is sufficient to link the observed increase of land surface temperature to the expansion of oil palm and other cash crops as the observed temperature increase at provincial level is in line with the observed temperature differences across land use type and the expansion of oil palm and cash crops over the last two and a half decades in the Jambi province. The area cultivated with oil palm grew faster than the area cultivated with rubber plantations between 1990 and 2011 (Clough et al. 2016). The title of our article is now: "Expansion of oil palm and other cash crops cause an increase of the surface temperature of the Jambi province in Indonesia"*

**III. Anonymous Referee #1**

General: The authors investigate the effect of land cover change (from forests to 'other' and mostly oil palm plantations) on regional land surface temperatures. They use remote sensing to determine LST (and albedo, NDVI, and ET). They conclude that conversion from forests had led to a ∼ 1 degree C temperature (positive) change after accounting for albedo. They also conclude that this is a positive feedback to climate warming.

**1. I only suggest some minor edits and (if the authors can) and expansion of the discussion of what these LST changes might translate to in the atmosphere?**
*The reviewer brings up an important issue of land surface – atmosphere feedback. A recent study by Tölle et al. (2016)[#] showed for SE Asia as a whole that land use change at large scale impacts the boundary layer structure, cloud-cover regime and other aspects of local and regional weather and climate. Particularly, land clearings can amplify the response to climatic extreme events such as El Nino Southern Oscillation. Analyzing this kind of effects require however a regional climate model, which is beyond the scope of our study. We now discuss this aspect in the manuscript and added the following sentence:*
*Line 773: "A recent study by Tölle et al. (2017) showed for SE Asia that land use change at large scale may increase not only surface temperature but also impact other aspects of local and regional weather and climate occurring also in regimes remote from the original landscape disturbance. Land clearings can amplify the response to climatic extreme events such as El Nino Southern Oscillation."*

**Tölle, M. H., Engler, S., and Panitz, H. 2017: Impact of Abrupt Land Cover Changes by Tropical Deforestation on Southeast Asian Climate and Agriculture. *Journal of Climate*, **30(7)**, 2587 – 2600, doi: 10.1175/JCLI-D-16-0131.1.**

2. How much larger of a region will they affect? How would you determine this?
*This study focuses on local and effects at the provincial level. Estimating or predicting the effects at a larger regional scale also requires a regional climate model as used in the study of*

*Tölle et al. (2017). Tölle et al. (2017) show that the effects of land use changes occur in remote regions other than where the land use changes occur. These effects are caused by the impacts the land cover change has on El Niño/La Niña episodes thereby enhancing wetter conditions in other regions, whereas in other regions wetter conditions decrease.*

Introduction: nicely written and I appreciate the well thought out definitions.

3. Line 96: missing an "as" after "such".
*We added the missing word as suggested.*

4. Methods: Could you describe the study sites in a little more detail (rather than the reference Drescher).
*We added the following information:*
*L. 186: "Previously logged rainforests in the Jambi province have been converted into intensively managed agro-industrial production zones as well as into smallholder farms to grow cash crop trees of rubber (Hevea brasiliensis) and oil palm (Elaeis guineensis) or fast-growing tree species such as Acacia mangium for pulp production (Drescher et al., 2016). The area cultivated with oil palm grew faster than the area cultivated with rubber plantations between 1990 and 2011 (Clough et al. 2016)."*

5. ET calculations: I'm familiar with the use of satellite data for all of the variables except for ET. Did you compare ET with the tower sites? How well does it work?
I see that you added this to the supplement, but it would be nice to have a validation of this method explained in the main text.

*We have ET and LE estimates from eddy covariance measurements for two oil palm plantations in Jambi Province (young and mature oil palm plantation). Our SEBAL based LE estimates are within the variability range of LE measured from eddy covariance under similar meteorological conditions.*

*We added the following text: "The SEBAL based LE estimates are within the variability range of LE measurements using the eddy covariance technique under similar meteorological conditions (see SI)."*

6. Results: Line 405-406: Hot = red? And cool = Blue colors. Can you please specify this?
*In our description of the figure we added the matching colors as suggested. Line 405 – 406: "the hot areas (red) correspond to the known clear-cut areas, urban areas or other sparsely vegetated areas, the cooler areas (blue) correspond to vegetated areas such as forest, plantation forests and mature oil palm plantations."*

7. Discussion: Line 668: When I look at the figures, there also seems to be a high correlation between NDVI and ET (simply because the response pattern, the pattern of the changes, look very similar). Can you explain this? Is it because of the ET calculation?
*Yes, ET and NDVI are highly correlated on one hand because the NDVI is used in the calculation of ET. On the other hand, another input for ET is LST, which is calculated from the raw thermal band (L6). L6 and NDVI are also highly correlated ($r = -0.87$) (see table 2, Line 494 – 502) even though NDVI and L6 are derived/measured from independent satellite bands. Thus, it come as no surprise that there is a correlation between NDVI and ET.*

8. Line 763: "concurrent to" should be "concurrent with"

*We changed the sentence with the correct prepositions as suggested.*

9. Line 768: "governmental" should be "government"
*We changed this and used the correct word.*

Final remarks: This is a well-written, well-organized manuscript. I support publication in Biogeosciences.
*We thank the anonymous referee for reviewing the manuscript and for the suggestions to improve the manuscript.*

**IV. Anonymous Referee #2**

General comments: Sabajo et al. evaluates the impact of land use changes on land surface temperatures in Indonesia over the MODIS timespan (1999-2015). The study is well written and provides a good, long-term observational analysis clearly showing the impact of regional deforestation on increasing land surface temperature across an entire region.

*We thank the anonymous referee for reviewing the manuscript and for the suggestions to improve the manuscript.*

1. The only general comment I have is that it would be good to include a **seasonality analysis** showing how deforestation has changed land surface temperature in both wet and dry season. I know that satellite remote sensing is more challenging during the wet season, but I think evaluating the impact of land changes with seasonality would be useful. This could also highlight likely reductions in ET with land change (and shallower rooting zones) during the dry season. The dry season is also when heat impacts (including wildfires) could be more significant.

*We agree that a seasonality analysis might show differences between the wet and dry season. We now made a seasonality analysis. Overall, the relationships in the dry season are stronger than for the wet season as we have much more usable data during the dry season. We found significant differences between LST of the dry and wet season. At 10:30 am the LST increased 0.09 ± 0.02 °C per year during the dry season, while the increase during the wet season was lower (0.06 ± 0.02 °C per year) (Fig. S10.1). Around 1:30 pm the LST increased 0.08 ± 0.03 °C per year, against 0.03 ± 0.02 °C increase per year during the wet season. At 10:30 pm the LST increased 0.03 ± 0.01 °C per year in the dry season, compared to a LST increase of 0.02 ± 0.01 °C in the wet season. At 1:30 am, the LST increased 0.05 ± 0.02 °C in the dry season, while the LST during the wet season increased 0.05 ± 0.03 °C. The increase of the LST at 1:30 pm, 10:30 pm and 1:30 am in the wet season was not significant (p = 0.12, p = 0.06 and p= 0.11, respectively). The significant increase of the LST during the dry season at all 4 times of observations suggests that the warming is more pronounced during the dry season compared to the wet season, which is reasonable as we have more incoming radiation during the dry season. Nevertheless, we prefer to pool the data from the dry and the wet season in order to get more statistically robust relationships.*

*In our analysis of the MODIS LST data we have not come across anomalous LST that could be attributed to forest fires. This is caused by the mask we applied in selecting the best quality pixels which mostly also removed pixels covered by smoke. A seasonality analysis is not possible with Landsat data because there is not enough data.*

*We added the following sentence to the manuscript (line 755):*
*"We like to point out that our MODIS analysis has a larger proportion of data from the dry season compared from the wet season, as there were more cloud free conditions during the dry season. Thus, our reported warming effect reflects cloud free conditions. During cloudy conditions, particularly in the wet season, the warming effect is expected to be lower."*

*We also added the seasonal analysis to the supplementary information (S10).*

*We thank the anonymous referee for reviewing the manuscript and for the suggestions to improve the manuscript.*

[Figure]

**Fig. S10.1** (from manuscript, extended): Wet and dry season are separated.
Wet season: All months except June – September/October; Dry season: June – September/October (Meijide et al., 2017 & Drescher et al., 2016).
This figure has been added to the Supplementary Information (S10)

2. Specific comments: For Figure 1 (and text in the manuscript related to Figure 1), it might be good to describe the general atmospheric circulation for wet and dry seasons (are winds from the east or from the west). This would help the reader evaluate whether there are substantial land use changes upwind of the forest plots that are used as the baseline "control" to evaluate land surface temperature changes to due land use changes and not overall global climate change. *We include a wind rose from one of our reference meteorological stations in the area, (see Drescher et al., 2016), for data collected between October 2013 to May 2016. Based on the climate diagram for the region (obtained from data from 1991-2011) we considered as the dry season the months of June-September and the rest was considered as wet season. See the wind roses for the dry (left) and wet (right) seasons below:*

[Figure]

[Figure]

*During the dry season winds were predominantly from the SE, whereas during the wet season winds where predominantly from the NW. The SE vs. NW shift in wind directions is in line with the regional monsoonal circulation. The landscape in the lowland of Jambi province is, however, very patchy with small-scale mosaics of different land uses. While we cannot fully rule out that advection from upwind land use changes may play a role, but it seems unlikely to have a systematic bias given the typical patchiness of the landscape. Also warm air advection would mean that the "climate change" warming of the forested "control" site is overestimated, thus making the land-use change effect even larger.*

**V. List of major changes**

1. We have changed the title to: *"Expansion of oil palm and other cash crops cause an increase of the surface temperature of the Jambi province in Indonesia".* The first title was too general, the results apply to the Jambi province and not to the whole of Indonesia.

2. We removed the reference of Paterson et al. (2015). After re-reading we concluded that this reference was not correctly cited and misplaced.

3. We added a new reference of Tölle et al. (2017). This reference was a new publication that complemented our results with a modelling approach.

4. We added 1 section to the supporting information (S10). S10 contains a seasonality analysis as suggested by anonymous referee #2. We also add a short sentence in the discussion and refer to the S10 for the results of the analysis.

5. Equations 9 – 11 have been renumbered from 10 – 12, due to a mistake in the equation 9 (which was by accident numbered as 1, while in fact that had to be equation 9).

6. We changed figure 5: we only adjusted the legend and paid attention to the rounding of the numbers in the equations.

[revised manuscript text omitted]